# The Rasch Analysis Shows Poor Construct Validity and Low Reliability of the Quebec User Evaluation of Satisfaction with Assistive Technology 2.0 (QUEST 2.0) Questionnaire

**DOI:** 10.3390/ijerph20021036

**Published:** 2023-01-06

**Authors:** Antonio Caronni, Marina Ramella, Pietro Arcuri, Claudia Salatino, Lucia Pigini, Maurizio Saruggia, Chiara Folini, Stefano Scarano, Rosa Maria Converti

**Affiliations:** 1IRCCS Istituto Auxologico Italiano, Department of Neurorehabilitation Sciences, Ospedale San Luca, 20122 Milan, Italy; 2IRCCS Fondazione Don Carlo Gnocchi Onlus, 20148 Milan, Italy; 3Department of Biomedical Sciences for Health, Università Degli Studi di Milano, 20129 Milan, Italy

**Keywords:** assistive devices assessment, neurological rehabilitation, neurological disability, psychometrics, Rasch analysis, many facets model

## Abstract

This study aims to test the construct validity and reliability of the Quebec User Evaluation of Satisfaction with assistive Technology 2.0 (QUEST)–device, an eight-item questionnaire for measuring satisfaction with assistive devices. We collected 250 questionnaires from 79 patients and 32 caregivers. One QUEST was completed for each assistive device. Five assistive device types were included. QUEST was tested with the Rasch analysis (Many-Facet Rating Scale Model: persons, items, and device type). Most patients were affected by neurological disabilities, and most questionnaires were about mobility devices. All items fitted the Rasch model (InfitMS range: 0.88–1.1; OutfitMS: 0.84–1.28). However, the ceiling effect of the questionnaire was large (15/111 participants totalled the maximum score), its targeting poor (respondents mean measure: 1.90 logits), and its reliability was 0.71. The device classes had different calibrations (range: −1.18 to 1.26 logits), and item 3 functioned differently in patients and caregivers. QUEST satisfaction measures have low reliability and weak construct validity. Lacking invariance, the QUEST total score is unsuitable for comparing the satisfaction levels of users of different device types. The differential item functioning suggests that the QUEST could also be problematic for comparing satisfaction in patients and caregivers.

## 1. Introduction

Assistive devices are expressly provided to improve independence and participation in people with activity limitations and participation restrictions [1]. Nevertheless, about one patient out of three no longer uses their assistive device one year after delivery [2].

Several reasons can lead to the abandonment of assistive devices, including patients’ improvement or worsening [3]. However, part of this abandonment is due to dissatisfaction with the device [4] and the device’s failure to meet patients’ expectations [2,5]. In broader terms, the relationship between satisfaction and use of devices is an aspect of the relationship between patients’ satisfaction and clinical outcomes [6].

Measuring satisfaction with assistive devices is thus essential. The Quebec User Evaluation of Satisfaction with assistive Technology (QUEST) is a questionnaire developed for measuring patients’ satisfaction with assistive devices [7]. Later, a short form, i.e., QUEST 2.0, was obtained from QUEST [8]. QUEST questionnaires have been used to evaluate patients’ satisfaction with different assistive devices [9], ranging from wheelchair seating and positioning to communication devices. One of the declared uses of the QUEST questionnaires is to compare the satisfaction of users of different devices [8]. Moreover, QUEST questionnaires have also been used to measure the caregiver’s satisfaction with the assistive device [10,11].

QUEST and QUEST 2.0 were developed with statistical techniques from the Classical Test Theory (CTT). CTT is the oldest set of statistics for questionnaire development and assessment and is probably still the most used for this purpose. According to the CTT, the total scores of the QUEST questionnaires have good reliability and validity [12]. 

However, questionnaire scores are not measures [13]. The most striking evidence of this is that a change in the questionnaire score from, for example, 5 to 6 does not necessarily imply the same increase in the quantity of the variable of interest (e.g., satisfaction) as a change from 12 to 13 (a 1 ≠ 1 paradox, precisely). In other words, scores lack linearity, an actual property of measures. Instead, questionnaire scores are just counts of observed events, *“essential for the construction of measures, but not yet measures”* [14].

The Rasch analysis assesses whether participants’ scores on questionnaire items satisfy a set of measurement axioms. If this occurs, questionnaires’ *ordinal scores* can be turned *into interval measures*, i.e., measures of the type of body temperature and blood pressure.

The advantages of the measures returned by the Rasch analysis over raw scores are clear. For example, versatile parametric statistics, including tests of significance (e.g., linear regression) and effect sizes (e.g., Cohen’s d), can be calculated on interval measures but not on ordinal scores. It has also been shown that measures from the Rasch analysis work better than questionnaires’ raw scores in practice, not just words [15]. High-quality measures, such as those validated with the Rasch analysis, increase the chance of making the right decision about patients [16]. Therefore, these measures are preferable to ordinal scores for clinicians and researchers.

Only two studies have used the Rasch analysis to assess QUEST 2.0 [17,18]. According to them, QUEST 2.0 was a good tool for measuring patients’ satisfaction in the Rasch measurement framework. However, QUEST 2.0 was administered only to orthosis [17] and prosthesis [18] users. Moreover, both works are from the same research group and evaluated only the Arabic version of the questionnaire.

As reported above, QUEST 2.0 is often used to measure and compare satisfaction in users of different assistive devices [8,9]. To the best of our knowledge, the Rasch analysis has never been used to evaluate the QUEST 2.0 psychometric properties as a generic measure of satisfaction. On these bases, the current work used the Rasch analysis to assess the construct validity and reliability of the QUEST 2.0 questionnaire in a variegated sample of assistive device users.

## 2. Materials and Methods

This observational cross-sectional study is part of an ongoing study to evaluate assistive devices’ effectiveness in people with chronic disabilities (Italian Ministry of Health-Ricerca Corrente, IRCCS Fondazione Don Gnocchi, Linea 4, project title: “Outcomes of Mobility Assistive Technology in rehabilitation pathways”). 

In the period between September 2016 and September 2021, the QUEST 2.0 questionnaire was administered to consecutive patients recruited according to the following inclusion criteria: (i) disability caused by a neurological or an orthopaedic disease; (ii) current use of at least an assistive device, (iii) age > 18 years, and (iv) disability duration ≥ one year. Patients’ exclusion criteria were: (i) inability to understand the questions in the questionnaire or to elaborate an answer and (ii) significant cognitive impairment, as gathered from the patient’s clinical assessment. 

If the patient had a relative identified as the primary caregiver, this caregiver could participate in the study and the QUEST was administered to them. Caregivers were recruited when they were assistive device users (e.g., they used an electrically powered bed when assisting their relative). No questionnaire was administered to them if they did not use the device themselves. Similar to patients, caregivers were included if they were over 18 years old and excluded if unable to understand questions in the questionnaire or elaborate an answer (e.g., poor proficiency with the Italian language).

Participants were outpatients referred to the IRCCS Fondazione Don Gnocchi, a rehabilitation facility in Milan. The sample represented a typical sample of chronic users of assistive devices accessing an Italian rehabilitation facility for device evaluation and prescription (http://portale.siva.it/en-GB/home/sivaCenters (accessed on 2 January 2023)) [19]. 

All patients were assessed by a physiatrist (RMC or MR) for the study’s recruitment. Regarding demographic information, age, gender, and the type of assistive device were gathered. In addition to these, the research staff also collected patients’ diagnoses from clinical records. However, because of different diagnostic criteria, with some patients receiving a syndrome-level diagnosis (e.g., tetraparesis) and others a disease diagnosis (e.g., multiple sclerosis), information about the diagnosis was used only to give a general picture of the patient’s sample. 

In line with [20], assistive devices were grouped as follows: home furnishings (e.g., bed, mattress), communication aids, aids for personal mobility (e.g., wheelchairs, powered wheelchairs, rollators), and lower limb prostheses. Moreover, the category “seating aids” (e.g., seat cushions) was added to this classification [21]. 

Patients were asked to read and fill out the QUEST 2.0 by themselves whenever possible. In the case of severe motor disability (e.g., tetraplegia), the questionnaire was filled out by one of the researchers who completed the patient’s clinical assessment. Caregivers were explicitly asked to fill the QUEST 2.0 by thinking about how satisfied they were with the assistive device (and not by considering how much they believed the patient was satisfied with it). 

The ethical committee of the IRCCS Fondazione Don Gnocchi (section of the Comitato Etico IRCCS Regione Lombardia) approved the study (protocol number: 10_16/04/2020), and participants and caregivers gave their written informed consent to participate in it.

### 2.1. The QUEST 2.0

The original version of the QUEST [7] consisted of 27 polytomous items scored in six categories. In a later study, QUEST developers simplified the questionnaire by reducing the items’ categories and the total number of items. As a result, QUEST 2.0 was obtained [8], a short form consisting of 12 items scored on 5 categories, ranging from 1 (“not satisfied at all”) to 5 (“very satisfied”). 

QUEST 2.0 items are arranged into two domains. The first (eight items) evaluates the patient’s satisfaction with the assistive device (QUEST 2.0–device), while the second (four items) assesses the patient’s satisfaction with the supply service (QUEST 2.0–service). The device domain asks how the patient is satisfied with eight features of the assistive device (“How satisfied are you with…”). From 1 to 8, these are as follows: dimensions, weight, easiness in adjusting, safety, durability, easiness of use, comfort, and effectiveness. The service domain tests the users’ satisfaction with the service delivery program, repairs and servicing, professional services, and follow-up services. 

The bidimensionality of QUEST 2.0 has been repeatedly confirmed. For this reason, it is customary to administer and analyse each of the two subscales separately (e.g., [17]). The current work aims to investigate the measurement of satisfaction with assistive devices. Hence, only the QUEST 2.0–device is analysed. 

One QUEST 2.0 for each assistive device should be collected [7]. Thus, multiple QUEST questionnaires are collected when a patient receives several devices, which is the case in the current study.

### 2.2. Rasch Analysis: Steps of Analysis

Rasch analysis was used to assess the reliability and construct validity of the QUEST 2.0–device. 

When questionnaires’ data comply with the model of Rasch, interval measures can be obtained from the questionnaires’ ordinal scores. “To comply with the model” means that the person’s scores to items verify the following assumptions: (i) items’ categories are ordered, (ii) items’ scores fit the model of Rasch, and (iii) the questionnaire score is unidimensional.

The Rasch analysis consists of different stages, each evaluating a distinct characteristic of the questionnaire, which will be outlined briefly below. 

More details on the analysis can be found in Appendix A from [22]. This supplement, addressed to clinicians interested in the Rasch analysis, sparsely uses statistics and mathematical notation, instead presenting the analysis more in a “qualitative” way. 

#### 2.2.1. Items Categories and Andrich Thresholds

The categories’ order can be checked by calculating the mean measure of the participants who responded in each category. Categories are ordered when there is a monotonic relationship between the category’s numeral and the corresponding mean measure, i.e., when the higher the item score, the higher the participants’ mean measure (and hence the quantity of the variable). 

Andrich thresholds are also commonly inspected when assessing the categories’ functioning. An Andrich threshold is the point along the line representing the construct (e.g., satisfaction) at which the chance of being scored in one of two adjacent categories (e.g., category 1 and 2) is the same. According to some authors, well-designed item categories have ordered Andrich thresholds [23].

#### 2.2.2. Fit to the Model of Rasch

In its original formulation, the Rasch model assumes that: (i) the probability of passing an item only depends on the difference between the item’s difficulty and the subject’s ability, and (ii) the shape of this relationship is that of the logistic function. 

Initially developed in education, the model can be easily adapted to measure medical and psychological constructs, such as satisfaction. When applied to the QUEST 2.0, items and respondents are aligned along the satisfaction construct. Therefore, the model is read as follows: “the probability of affirming an item depends on the difference between how a user is satisfied (i.e., the user satisfaction level) and how hard it is to satisfy the device feature the item describes (i.e., the satisfaction level needed to endorse the item)”.

Fit to the model is given by the mean square (MNSQ) and the z-standardised (ZSTD) statistics. In plain terms, MNSQ quantifies the departure size from the model’s prediction, while ZSTD gives the statistical significance of this departure. 

Two versions of the MNSQ and ZSTD are commonly computed. “outfit” MNSQ and ZSTD are sensitive to outliers, while inliers rather than outliers influence “infit” MNSQ and ZSTD. 

In statistical terms, infit and outfit MSNQ are calculated from the squared standardised residuals, similar to the conventional chi-square statistic (the outfit MNSQ is Pearson’s chi-square divided by its degrees of freedom, actually). The ZSTD statistic can be considered a t-test of the null hypothesis: “data do not depart from the model”.

As customary, an item was considered not to fit (i.e., to misfit) the Rasch model if its infit *or* outfit MNSQ values were outside the 0.5 to 1.5 range *and* the corresponding ZSTD were less than −1.96 or larger than 1.96. In other words, a large *and* significant model departure is needed to flag an item as misfitting the model. 

#### 2.2.3. Testing Unidimensionality: The Principal Component Analysis of the Model’s Residuals

The Rasch measurement model assumes that questionnaires are unidimensional, i.e., their scores depend on a single variable. In the Rasch analysis, it is customary to calculate a principal component analysis (PCA) of the model’s residuals to evaluate if a questionnaire is unidimensional. 

If a questionnaire is truly unidimensional, when the Rasch dimension is “peeled off” from the data, what remains (i.e., the model’s residuals) is random noise (i.e., uncorrelated variance) [24]. On the contrary, if one or more principal components with eigenvalue > 2.0 are found (i.e., if the PCA points out strong covariance among the residuals), one or more additional variables influence the items’ scores, and the questionnaire is considered multidimensional.

Running the PCA of the residuals is not straightforward for the current analysis as it is for a typical two facets, participants and items, dataset. The present study collected more than one questionnaire from most participants, and participants contributed to the dataset with a different number of questionnaires.

Only data from users of mobility devices were used for the PCA (one dataset row per participant), precisely to avoid people with more questionnaires weighted more in the analysis. 

#### 2.2.4. Differential Item Functioning

In the current study, differential item functioning (DIF) tests if scores to an item are the same in participants *with the same level of satisfaction* but belonging to different groups (e.g., males vs. females). If this is not the case, another variable, in addition to satisfaction, affects the item’s score (e.g., gender). Uniform DIF was tested here for the following variables: gender (males vs. females), age (<65 years vs. ≥65 years), and respondent type (patient vs, caregiver). These variables were chosen since it can be well-expected that satisfaction can have meanings that are subtly different in males and females, the young and elderly, and patients and caregivers, thus possibly causing differences in items’ functioning. 

For example, the weight of a wheelchair could be more crucial for an elderly than a young adult; therefore, satisfaction with the device weight (item 2) could be easier to endorse in the latter than in the former. Likewise, on this line of reasoning, satisfaction with the device’s comfort (item 7) could be easier to endorse for a caregiver than for a patient since, for example, patients spend several hours sitting in a wheelchair. 

Moreover, age and gender have already been assessed in DIF studies of satisfaction measures (e.g., [25,26]). In addition, DIF for satisfaction items and caregiver status has also been investigated (e.g., [27]). 

It should also be stressed that DIF should be preliminarily tested anytime it is interesting to compare the measured variable in different groups. In plain terms, if a researcher is interested in studying if device satisfaction is different in patients and caregivers, they should first investigate if the measurement instrument is invariant (i.e., its items have no DIF) for the respondent type. 

DIF was examined following the procedure developed by Linacre [28]. First, the calibrations of the QUEST 2.0 items (and their standard errors) were calculated in the opposite DIF groups (e.g., patients and caregivers). Next, a t-statistic was calculated from the items’ calibration and standard errors to assess the DIF significance. The following null hypothesis is tested in a DIF analysis: “the item’s calibration is not different in the two DIF groups“.

DIF is considered non-dismissible if: (i) the item’s calibrations are significantly different (*p* < 0.01) in the two groups of respondents, and (ii) this difference is > 0.5 logits. 

As a final note, it is worth noting that DIF tests the questionnaire’s dimensionality from a different perspective. For example, if an item has a DIF for gender, its score depends on two variables: The Rasch dimension *and* gender. However, it is customary to refer to the PCA as the dimensionality test, which will be done in this study. 

#### 2.2.5. The Item Map

Once it is shown that this model’s assumptions are met, and thus interval measures can be derived from ordinal scores, it is meaningful to assess the quality of these measures (i.e., their validity and reliability).

The items map, a powerful tool for assessing the construct validity of a questionnaire [29], is a graphic representation of the items’ calibration and persons’ measures along the line representing the construct of interest (i.e., satisfaction). As a note, “calibration” and “measure” have the same meaning in the Rasch measurement theory, but the former is used for items and the latter for persons. 

The simultaneous representation of items and persons makes it immediately apparent if a questionnaire is well-targeted to the participants’ sample. If this is the case, the participants’ mean measure is about symmetrically distributed around 0 logits, the items’ mean calibration by default. The questionnaire “targeting” refers to the difference between the mean persons’ measure and the mean item calibration.

The questionnaire ceiling (and floor) effect are also considered when assessing the questionnaire map, the former given by the number of respondents scoring the highest category in each answered item (and the latter scoring the lower one).

#### 2.2.6. Persons’ Reliability

In the framework of the Rasch analysis, the reliability is given by the persons’ reliability, an index analogous to Cronbach’s alpha, reflecting the precision of the persons’ measures.

In line with the classical definition of reliability, Rasch persons’ reliability is given by the ratio of the “true” variance of the persons’ measures to the variance of the observed persons’ measures.

Each Rasch measure is accompanied by a standard error representing the standard deviation of the measurement errors. The “true” persons’ measures variance is thus given by the difference between the variance of the observed measures and the mean of the squared standard errors of the persons’ measures (i.e., the variance of the error). In plain words, the “true” measures variability is the variability of measures after removing the variability due to the measurement error [30,31].

From persons’ reliability, it is possible to calculate the number of “strata”, the number of measures’ levels significantly different at a single subject level. For example, three strata indicate that a disability questionnaire can show a patient’s recovery from severe to moderate disability and eventually to mild [32,33]. Persons’ reliability should be at least 0.8, so the questionnaire distinguishes three strata [32]. 

### 2.3. Rasch Analysis: Which Model of the Rasch Family?

The Many-Facet Rating Scale model was used for the current analysis. The Rasch model [34] is only suitable for analysing dichotomous items in its original formulation. Next, the model was elaborated into the rating scale [35] and partial credit [36] to analyse polytomous items. To date, the “Rasch model” represents a family of measurement models, and the many-facet model [37] is one of its most recent additions.

The original Rasch model, the rating scale, and the partial credit models analyse data from the interaction of two facets: items and persons. The many-facet model extends them by adding (at least) a third facet (e.g., raters, occasions) [38]. In statistical terms, the many-facet model provides a convenient way to run the Rasch analysis when there are repeated measurements, which is why it has been used here. 

For the current study, a three-facets model was used: participants, items, and the type of assistive device. It is worth stressing that the third added facet does not modify how the analysis results are interpreted, starting with the items map. The many-facet model aligns patients, the QUEST 2.0–device items, and the type of assistive device along the same line, representing the satisfaction continuum (low to high). Patients scarcely and fully satisfied are placed on the low and high end of the line, respectively. Items and assistive devices patients are “easily” satisfied with are positioned on the low end. Conversely, items and assistive devices “difficult” to endorse are placed on the high end. 

The rating scale variant of the many-facet model was used here instead of the partial credit one for two main reasons. First, QUEST 2.0–device categories have the same numerals and descriptions in each questionnaire item. Hence, a rating scale analysis is totally legitimate. Second, for sample size considerations, we preferred to have robust estimates testing a simpler model with fewer parameters.

Facets version 3.84.0 was used for the primary analysis (Many-Facet Rating Scale model), and Winsteps version 5.2.5.2 for calculating the PCA of the model’s residuals. R [39] was used for additional analyses and graphics.

## 3. Results

The study included 250 questionnaires, 184 from 79 patients and 66 from 32 caregivers. Most questionnaires were about mobility devices, followed by seating aids (Table 1). Two or more questionnaires were collected from 87 participants. At most, seven questionnaires were collected from a single participant.

A few missing items occurred (about 3% of the size of the expected data matrix), more frequently on items 2 and 3, while no missing data occurred for item 1 (Appendix A).

The patients’ sample (mean age: 66.8 years; SD: 18.9 years; 62 elderlies; 45 males) had heterogeneous diagnoses, primarily neurological (69 patients). At a syndrome level, 20 had tetraparesis, 19 had hemiparesis, and 11 had parkinsonism. The three most common classes of neurological diseases were: stroke and other cerebrovascular accidents (14 patients), multiple sclerosis (8 patients), and amyotrophic lateral sclerosis and other diseases of the motorneurones and muscle (7 patients). 

The study also recruited 16 patients with a disability secondary to an orthopaedic impairment, including nine patients with lower limb amputation, five of whom were users of lower limb prostheses. The remaining patients had a disability of cardiorespiratory or multifactorial origin (e.g., gait impairment in the oldest old).

### Rasch Analysis of the QUEST 2.0–Device: A Combined Study of Patients and Caregivers

The Many-Facet Rating Scale model showed that QUEST items had ordered categories and Andrich thresholds (Table 2). However, category 2, “not very satisfied”, emerged on the latent variable continuum for a short tract (from −0.34 to −0.28 logits, i.e., 0.06 logits). It is also noteworthy that the categories’ distribution frequency was skewed, with categories 1 and 2 rarely chosen. Additional details on the categories can be found in Appendix A. 

All eight QUEST 2.0–device items fitted the Rasch model (infit MNSQ range: 0.88 to 1.10; outfit MNSQ range: 0.84 to 1.28; Table 3). However, regarding the device classes, questionnaires about furnishings showed large and significant outfit statistics (MNSQ: 1.59; ZSTD: 2.45), indicating a poor fit to the model. 

Figure 1 shows the map of QUEST 2.0–device.

Patients’ measures, items’, and devices’ calibrations are referred to an interval scale with one logit as the measurement unit and centred on 0, i.e., the items’ mean calibration. Low negative values indicate low satisfaction, and high positive values high satisfaction. Hence, low negative values indicate participants are poorly satisfied, and items and devices are easily endorsed (i.e., a low satisfaction level is enough to endorse the item/device). On the contrary, high positive values indicate highly satisfied participants, and items and devices are difficult to endorse.

The lowest satisfaction level is flagged by item 5 “durability” (−0.39 logit). In contrast, one must be highly satisfied to endorse item 2 “weight” (0.36 logit). In other words, “durability” is the device feature that is easier to be satisfied with, while the device “weight” is the most difficult to satisfy. 

QUEST 2.0–device is affected by an apparent ceiling effect. About 16% of participants (15 out of 111) scored the maximum category in each questionnaire’s items. No participant totalled the questionnaire’s minimum score. Participants were poorly centred on the map, indicating low questionnaire targeting. The participant’s mean measure was 1.90 logits (SD: 1.47 logits), definitively above the items’ mean calibration (extreme high scores removed: participants mean measure: 1.48 logits; SD: 1.06 logits). 

The map also shows that the assistive devices have different calibrations, with communication aids measuring low (−1.18 logits) and prostheses measuring high (1.26 logits; Figure 1D). This result indicates that the items’ calibrations differ for users of different device types. 

Regarding dimensionality, the eigenvalue of the first principal component from the PCA of the model’s residuals was 1.78. The variance explained by Rasch measures was just 35.3% of the total variance of the dataset.

DIF was found for respondents, with item 3 more difficult to endorse for patients than caregivers (contrast: 0.52 logit, SE: 0.19 logit; *p* = 0.006). No DIF was found for age and gender. The full results of the DIF analysis are reported in Appendix A.

Person reliability was 0.71, and the number of strata was 2.42. When the 15 extreme participants were removed from the analysis, reliability rose to 0.81 and strata to 3.13.

## 4. Discussion

The Rasch analysis showed that, even if the QUEST 2.0–device questionnaire has some psychometric strengths, it suffers some critical weaknesses that make its measures to be poor satisfaction measures.

Among the questionnaire’s strengths, it should be stressed that all eight items of the QUEST 2.0–device fit the model of Rasch. In addition, QUEST 2.0–device returns a unidimensional measure of satisfaction with the assistive devices. However, the ceiling effect of the questionnaire is high, its targeting is poor (a finding indicating poor construct validity), and its reliability is low. Moreover, QUEST 2.0–device lacks measurement invariance since its item calibration depends on the assistive device type. 

### 4.1. The Item Map: High Ceiling Effect of the QUEST 2.0–Device

Its ceiling effect is likely the most apparent issue of the QUEST–device. The ceiling effect is familiar with satisfaction questionnaires [6,40]. However, a significant ceiling effect raises both conceptual and methodological issues.

First, it is unlikely that many patients are satisfied precisely to the same amount [7]. In addition, it is unlikely that patients are completely satisfied (i.e., there is no room for improvement) [41]. 

The ceiling effect is a clear threat to responsiveness, that is, the ability of measures to highlight the genuine improvement (or worsening) of the patient [42,43,44]. In plain words, patients’ progress passes unseen if their measures are already at their maximum. This issue is not just a theoretical one. For example, it has been estimated that clinical improvement can pass undetected even in one patient out of five because of the ceiling effect [45]. 

The ceiling effect can also cause difficulties with data analysis [46]. In strict statistical terms, by causing low response variability [47], a significant ceiling effect can affect the correlation analyses, thus masking the association between satisfaction and other variables [48]. A prominent ceiling effect in satisfaction measures has been considered to hide the association between patients’ clinical improvement and treatment satisfaction [49]. 

The ceiling effect of satisfaction questionnaires can have several explanations, not necessarily linked to genuine satisfaction. For example, in front of a healthcare provider, patients could give top scores to a questionnaire measuring satisfaction for different reasons, such as respect, deference, or generosity [48]. The social desirability bias, the tendency to underreport undesirable symptoms, attitudes, and behaviours that may be regarded as unfriendly or unwelcome, could also play a role in causing the ceiling effect and the limited variability of satisfaction questionnaires [41]. A long-standing personal relationship with the staff, common between patients with disability and their clinicians, is also supposed to favour satisfaction [50]. 

### 4.2. The Item Map: Poor Targeting of the QUEST 2.0–Device

The item map also highlights the poor targeting of the questionnaire to patients’ satisfaction. 

The map clearly shows that the participants’ mean measure is between 1 and 2 logits, definitely above the items’ mean measure (0 logits). According to Rasch modelling, patients whose total scores are close to the questionnaire’s maximum (or minimum) are poorly measured (i.e., questionnaires work poorly to extremes). In fact, for very high (or low) scores, the change of one point of the total score indicates significant changes in the person’s measure. Therefore, the precision of the measurement is low [51]. 

In addition, from the map, it is also apparent that several participants measure 1–4 logits where no item (and no Andrich threshold) is present. Therefore, the QUEST 2.0–device items poorly investigate a wide range of satisfaction, indicating that the questionnaire does not correctly assess this construct region. As a result, the construct validity of the QUEST 2.0–device is eventually low [29]. 

### 4.3. Person Reliability of the QUEST 2.0–Device Is Low

The analysis also pointed out the poor reliability of the questionnaire. As a result, the QUEST 2.0–device can only distinguish two strata of satisfaction. This finding means that the QUEST 2.0–device only discriminates patients with low and high satisfaction when measured at a single subject level. Furthermore, from another point, the QUEST 2.0–device detects the patient’s improvement only if their satisfaction increases from low to high. 

As mentioned above, poor measurement precision is somehow expected in the presence of a significant ceiling effect. Conversely, a more negligible ceiling effect would mean more score variability and more information to differentiate patients [48].

Splitting patients into two strata is like dichotomising patients between “satisfied” and “unsatisfied” patients based on whether their score is maximal or not [48]. This analysis is often done with satisfaction questionnaires, again because of the large ceiling effect. However, the dichotomisation of questionnaires’ scores is commonly considered a weak analysis solution [52]. 

### 4.4. Additional Psychometric Flaws of the QUEST 2.0–Device: Differential Test Functioning and Differential Item Functioning

The QUEST 2.0–device map shows that the different assistive devices have different calibrations, indicating that comparing the QUEST 2.0–device total raw scores of users of various assistive devices could lead to wrong conclusions. 

Consider two students grading the same on history. However, since different teachers assessed them, the teachers’ severity/leniency should be considered for a fair comparison of the two students. The same happens with the QUEST 2.0–device score and the assistive device class. For example, the item map indicates that it is easier to be satisfied with communication aids than prostheses. So, given a prosthesis user and a communication aid user totalling the same QUEST 2.0–device score, the prosthesis user’s satisfaction measure is higher than that of the communication aid user. In these terms, the many-facet analysis points out that the QUEST 2.0–device is not sample-independent (a fundamental feature of Rasch measurement [53]) since its scores function differently for users of different devices. In a sense, it can be proposed that the shift of the items’ calibration with the class of assistive device indicates that the QUEST 2.0–device is affected by differential test functioning (DTF) [54]. Appendix A further develops this point. 

Another piece of evidence that the questionnaire works differently for different devices comes from the data-model fit analysis. Even if the QUEST 2.0–device items fit the Rasch model overall, this does not seem to be the case for the questionnaires about furnishing devices, which weakly follow the model expectations. The analysis showed that items’ scores are relatively far from the model’s prediction when the QUEST 2.0–device is administered to users of furnishings. Strictly speaking, no trustworthy measure can be derived from the questionnaire score when data do not fit the model. 

The DIF analysis further evidences the lack of measurement invariance of the questionnaire. Item 3 was affected by DIF, being easier to endorse for caregivers than for patients. Consider a caregiver and a patient with the same level of satisfaction overall. DIF implies that the caregiver’s score on item 3 is higher than that of the patient. Since the two participants enjoy the same level of satisfaction, something else affects this item’s score independently from satisfaction. 

The artefact caused by the DIF on measurements is two-way. As just illustrated, DIF not only causes a patient and a caregiver who measure the same to score differently on the same item, but also patients and caregivers with the same score can have different satisfaction levels (and hence satisfaction measures) because of DIF. The measurement problem is thus overt. 

This DIF of item 3 has a plausible explanation. Item 3 asks about the level of satisfaction in adjusting the parts of the assistive device. Being disabled, adjusting (e.g., fastening) their assistive device can be harder for patients than caregivers; therefore, they are less likely satisfied with this specific aspect of the assistive device. In this vignette, hand dexterity impairment could be the additional variable affecting the score of item 3 (the higher, the lower the item’s score). 

Once DIF is found, procedures are available in the Rasch analysis for removing the measurement artefact caused by it [55]. In the statistical jargon and regarding the current findings, the Rasch model can control for the difference in the item 3 calibration for patients versus caregivers. Similarly, the many-facet model can control for differences between the device classes [56]. 

As explained above, ordinal scores are unfair for comparing respondents of other classes in the case of DIF or DTF. However, fair measures can be obtained from these unfair scores with appropriate procedures [37]. Nevertheless, DIF and DTF solving were not implemented because of the questionnaire’s other measurement weaknesses and the introductory nature of the current study (see the section on study’s limitations).

Regarding the assessment of the QUEST 2.0–device in patients and caregivers, it is worth pointing out that measuring the caregivers’ satisfaction with devices is as meaningful as measuring that of the patients since, in several instances, caregivers are also users of the assistive device (e.g., [57]). Comparing the questionnaire functioning in patients and caregivers, as done with the DIF analysis discussed above, is thus not only entirely appropriate but also highly recommended. A questionnaire DIF-free for respondent’s type is needed anytime interested in comparing the patients’ and caregivers’ satisfaction [11]. 

### 4.5. Rasch Analysis for Questionnaires’ Refinement and Development

Rasch analysis is a valuable tool, especially for developing and improving questionnaires.

There have been various attempts to reduce the ceiling effect of existing questionnaires. However, modifying a satisfaction questionnaire to reduce its ceiling effect is likely not as easy as it sounds. A solution often adopted to reduce the ceiling effect is rescoring the items into more positive than negative categories [58]. 

Regarding the QUEST 2.0–device, the current analysis showed that the thresholds between category 2, “not very satisfied”, and category 3 “, more or less satisfied”, were very close; therefore, category 2 emerged for a very narrow range of satisfaction. According to some authors [23], in this case, categories could be collapsed, and a four-category structure could work better than the original one of five. Using this line of reasoning, the new categories could be relabelled from 1 to 4 as follows: “I’m not satisfied”, “I’m a bit satisfied”, “I’m very satisfied”, and “I couldn’t be more satisfied” [59]. Of course, this change in the labelling of the categories requires new experimental testing. 

The item map also gives precious hints for questionnaires’ refinement. New items probing high satisfaction levels should be included in the QUEST 2.0–device to improve its targeting and ceiling effect. In plain words, ticks (i.e., thresholds) should be added to the positive part of the ruler (i.e., the item map, in the range from 1 to 4 logits) to increase the precision in the measurement of people with high satisfaction. The QUEST 2.0–device map indicates that the device’s features that are more difficult to endorse are the device’s weight (item 2) and the ease of adjusting the device’s parts (item 3). Device characteristics that could be even more challenging to satisfy could include, for example, the device’s aesthetic appearance. 

In searching for new items, it should be remembered that patients’ satisfaction reflects the cognitive evaluation and emotional reaction to the care received [60]. When they fill out a satisfaction questionnaire, patients recall (more or less consciously) their assessment and their emotive reaction elicited by the device they received [50]. Therefore, items assessing emotional aspects linked to the device could also be tested (e.g., “How much are you satisfied with the way this wheelchair makes you feel comfortable among strangers?”).

The need to modify the QUEST 2.0–device items has been already put forward. It was recognised that features relevant to some specific assistive devices, such as “speed” for powered wheelchairs users, are absent in the QUEST 2.0–device [8]. Of course, this would lead to device-specific questionnaire items. QUEST 2.0–device items could also be better detailed. For example, ease of use (item 6) can be split into the ease of donning and doffing the assistive device (e.g., prostheses). Likewise, satisfaction with dimensions (item 1) could be detailed as satisfaction with the height and satisfaction with width (e.g., wheelchair). 

### 4.6. Previous Works Using the Rasch Analysis to Assess QUEST 2.0

To our knowledge, the current work is the third study in which the Rasch analysis was used for assessing QUEST 2.0. However, the two previous works [17,18] have tested QUEST 2.0 only in orthosis and prosthesis users. In addition, only the Arabic version has been evaluated. 

The results reported here are in line with those from those previous studies. For example, they also found ordered categories and Andrich thresholds [17,18], with category 2 emerging for a narrow range of measures [18]. In addition, the fit of items to the model was satisfactory [17,18], and the PCA of the QUEST 2.0–device showed unidimensionality [18].

Among the questionnaire’s cons, its ceiling effect was pointed out [18], and both studies highlighted reduced questionnaire targeting [17,18]. The need for new items probing high satisfaction to improve the measurement features of the questionnaire has also been discussed [17].

Regarding the questionnaire’s targeting, it is noteworthy that the value found here is much worse than that reported by [17] and [18]. However, the many-facet modelling shows that these findings are not contradictory but just a consequence of the different devices’ calibration. As reported above, [17] and [18] only administered QUEST 2.0 to orthosis and prosthesis users, with a questionnaire targeting (i.e., the participants’ mean measures) of 0.67 and 0.69 logits, respectively. Instead, the mean measure of the prosthesis users recruited here was 1.97 logits.

Concerning this, it should be remembered that: i) in Rasch analysis, participants’ measures are referenced to the items’ mean calibration (0 logits), and ii) the prostheses calibration (1.26 logits; Table 3) is higher than the other devices.

When the mean measure of the prosthesis users recruited here is referenced to the prostheses calibration (i.e., the items mean calibration for prostheses), as done in [17] and [18], their mean measure (1.97 − 1.26 = 0.71 logits) is superimposable to that of the former works. 

Regarding again the similarities among the three studies, the narrow range items’ calibrations of the present analysis (0.75 logits; from −0.39 to 0.36 logits) is comparable to that reported by [18] (0.73 logits; from −0.33 to 0.40 logits). On the other hand, the item range found by [17] was larger (1.15 logits range), but this is because the calibration of item 7 (0.73 logits) was higher than that of the others. Intriguingly, the remaining items ranged between −0.42 and 0.35 (0.77 logits).

Regarding the analysis of the questionnaire invariance, no DIF was found for several variables (e.g. age, gender, country, patients’ clinical features) [18], and a stricter comparison of the questionnaire functioning in users of different devices was encouraged [17]. 

### 4.7. Rasch Models and Rasch Measures from the Perspective of Statistical Modelling

The database analysed here originates from the interaction of three facets: persons, questionnaire items, and assistive device type. The Many-Facet Rating Scale model was used since it can return actual Rasch measures from measurement situations as complex as the current one or even more complex (e.g., time series). 

However, it must be pointed out that the many-facet model is just one possible solution to the measurement problem faced here and that, as is often the case in statistics, the same dataset can be analysed with different statistical tools. 

The Rasch analysis is an umbrella term that embraces a family of measurement models. To date, new models have appeared complementing the “historical” ones, i.e., the original Rasch model, the Andrich rating scale, the Masters partial credit, and the Linacre many-facet models [61]. 

Intriguingly, the Rasch model can be understood as a generalised linear mixed model (GLMM) [62]. Without going into details, it would correspond to a random person–fixed items model, i.e., a model with respondents as random effects and items as fixed effects, with the logit function as the link function and the data following the Bernoulli distribution [63,64]. 

As with any linear model, GLMM Rasch models can flexibly accommodate covariates, such as the type of assistive device, thus allowing, for example, the analysis of the database of the current study. 

In addition, new Rasch models have been developed in the GLMM framework. One is the random persons–random items model, which considers the selected items as randomly chosen from a universe of possible items and does not put a specific interest in the sampled items [63,65]. This idea is entirely in line with a key idea of the Rasch analysis: that person measurements should be item-free (i.e. independent from the particular items used for measuring them) and that item calibrations should be person-free [66]. Moreover, Rasch models with random items allow for robust DIF analyses [63].

An emphasis has been put in the current work that one of the strengths of Rasch-consistent items is that their ordinal scores can be turned into measures. 

Measures from the Rasch analysis have significant advantages. 

First, theoretically, they are “true” measures precisely like those from mathematics, physics, and chemistry. This point is a major one from a philosophy of science perspective, given the well-known discussion about the possibility of measuring latent constructs [67]. Second, Rasch measures have been shown to work better than raw scores in empirical studies, not just in theory [15]. Finally, regarding the data analysis, Rasch measures give access to well-established statistical tools, such as linear regression, in the first place, which in strictly theoretical terms should be avoided on ordinal predictors and ordinal response variables. 

However, all this does not mean that ordinal scores and nominal categorisations are useless. On the contrary, we feel obliged to stress that even if these are not measures [13], they can provide information of the utmost importance. 

Here, it is sufficient to notice that medical diagnosis (i.e., nominal categorisations) and clinical (ordinal) scores (e.g., [68]) can be life-changing for patients. 

Finally, it is also worth stressing that, from a data analysis perspective, appropriate statistics are available for the rigorous analysis of ordinal data [69]. 

### 4.8. Study’s Limitations

Among the limitations of the current work, the most apparent is likely linked to DIF analysis. As already reported, the DIF analysis of the QUEST 2.0–device run here should be considered introductory. 

First, only three variables were evaluated (gender, age, and respondent type). Second, some scholars could consider the sample size of the current study unsuitable for a DIF analysis. Even if different authors have put forward a range of sample sizes for DIF assessment [28,70,71], some guidelines on Rasch analysis indicate that at least 200 participants per group are needed [72]. However, this rule of thumb gives rise to criticism: it forces to the medical field a rule of thumb used in education [71] and does not consider the many aims of the DIF assessment. 

The DIF analysis can have severe consequences on questionnaires’ development and refinement. When DIF is found, items showing DIF can be removed or undergo the “split procedure” to have DIF-amended item calibrations [73]. However, it has been noted that both procedures can weaken the questionnaire’s validity [55]. Therefore, a sample size allowing a “definitive” analysis is needed for such a profound modification of a measurement instrument. 

However, it should be considered that DIF assessment is also valuable to further test the independency of item calibrations from the persons’ sample, an actual requirement of Rasch measurement [52]. With this aim, smaller DIF samples can be used in introductory questionnaires’ evaluation [28].

It is also worth stressing about DIF that the consequences on measures of the DIF found here remain to be ascertained. DIF points out that *an item score* does not measure correctly. However, the *total questionnaire score* can be robust to the malfunctioning of a few items, and it is well-known that, even if present, DIF could be practically irrelevant at a whole questionnaire level [22]. 

The clinical profiling of patients needs to be improved. As explained in the Methods section, we have not been able to provide a better diagnostic picture of the patients recruited here since different diagnostic criteria were mixed in the patient’s clinical records. 

Again, with additional clinical information, the DIF analysis could be extended to assessing the questionnaire invariance in different diseases (neurological vs. orthopaedic conditions, in the first place) and various syndromes (e.g., paraparesis vs. tetraparesis). In this scenario, it would also be of interest to extend the DIF study to other clinical variables, such as the patient’s disability level (e.g., slightly vs. severely disabled patients) and cognitive impairment (e.g., present vs. absent). 

The majority of questionnaires collected in this study were from mobility device users. Users of communication aids and furnishings were much less represented, and prostheses users were the most underrepresented. This fact affected the estimates’ precision as indicated by standard errors, which, as expected, were larger for prosthesis users whose subsample consisted of less than ten respondents [74]. 

However, the confidence intervals of the device’s calibrations also indicate that these estimates are trustworthy despite the limited sample size. Keeping in mind that, in the Rasch measurement, differences between estimates should be > 0.5 logits to matter [75], the 95% confidence interval of the communication aids’ and furnishings’ calibrations is narrower than 0.5 logits, while that of the prostheses is just slightly above this threshold. 

In addition, after properly equating the item maps (see the previous section), the mean measures of the prostheses users are comparable to those of previous studies [18], pointing out that the prostheses measures here are likely trustworthy despite the significantly reduced sample size. 

Regarding the total number of participants for the primary analysis, we know it is not exceptionally high for psychometric research. However, it is noteworthy that 50 participants are enough for exploratory testing of a questionnaire made of polytomous items in the Rasch modelling framework [76]. Furthermore, a sample size of 100 participants returns item calibration estimates stable within a ± 0.5 logit range [76]. 

It is also noteworthy that the sample size of the current work is comparable to that of the previous QUEST 2.0 Rasch analyses (111 respondents vs. 100 in [17] and 183 in [18]) and that the current work collected the largest number of questionnaires (250) amongst the studies.

The availability of more questionnaires, managed adequately by the many-facet model, likely improved the estimation of the parameters, as indicated by the standard error of the item’s calibrations (median: 0.08 logits), which are virtually the same as that declared by [18] (median: 0.10 logits) and better than that reported by [17] (median: 0.15 logits).

Another limitation in this study deals with a possible selection/attrition bias. 

The participants were recruited in follow-up visits after receiving their assistive devices, which could have been delivered months or even years before the study enrolment. It cannot be excluded that some users received an assistive device but did not attend the follow-up visits. Consequently, enrolling these patients in the current study was not feasible. 

This eventuality could have caused a selection bias. For example, there is the possibility that the users lost to follow-up were those dissatisfied with their devices. Including these patients in the study would have reduced the QUEST ceiling effect and improved the questionnaire targeting.

However, it should be stressed that a central finding of the current study is that patients who are “so so” satisfied are poorly measured. This fact is immediately apparent from the participants’ map (Figure 1A), which, as discussed above, makes clear that QUEST 2.0 items badly investigate a wide range of satisfaction, thus poorly measuring people who are not completely satisfied with their device. 

## 5. Conclusions

The Rasch analysis highlighted some measurement flaws of the device domain of the QUEST 2.0 questionnaire. To summarise, QUEST 2.0–device suffers a severe ceiling effect, poor item targeting of the respondents’ sample, low reliability, and lack of measurement invariance. Regarding this last point, total scores of the QUEST 2.0–device should not be used to compare satisfaction in users of different device types. 

However, the current analysis also gives helpful cues for future questionnaire development. First, the eight items of the QUEST 2.0–device have a proper fit to the Rasch model overall and thus can work as the core of the improved questionnaire version. Second, the analysis suggests that four-category items could work better than the original five. Third, the item map allows a deeper understanding of the satisfaction with assistive devices and indicates the satisfaction level that the new questionnaire items should probe. 

Finally, suppose future studies confirm the lack of invariance of the questionnaire’s items for the assistive device type. In such a case, one must remember that measures from the many-facet model represent a formal solution to this problem [56], allowing a fair comparison of satisfaction in users of different device types [37].

## Figures and Tables

**Figure 1 ijerph-20-01036-f001:**
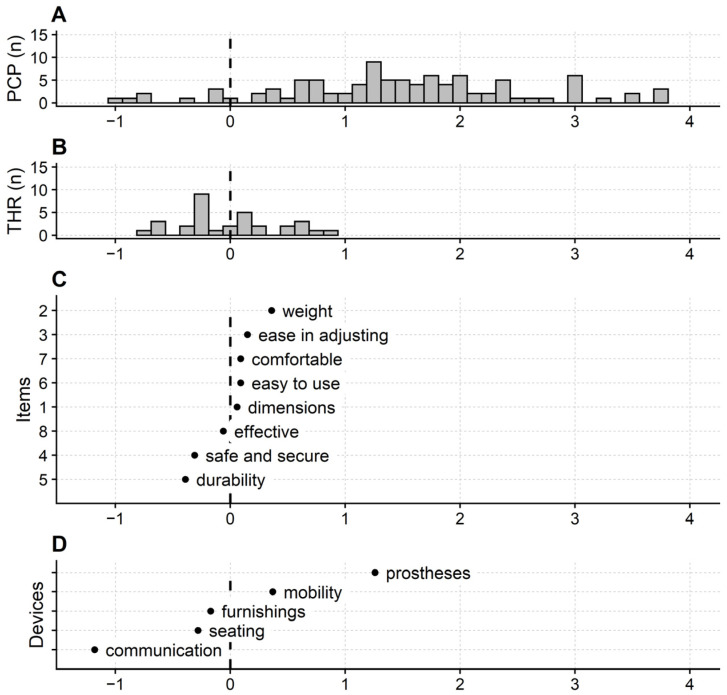
Questionnaire map of the QUEST 2.0–device. (**A**): distribution frequency of the patients’ measures. (**B**): distribution frequency of the items’ thresholds calibrations (Andrich thresholds). (**C**): items’ calibrations. (**D**): devices’ calibrations. Participants, items (with the Andrich thresholds), and devices are the first, second, and third facets of the Many-Facet Rating Scale model used for the Rasch analysis. PCP: participants; THR: thresholds; *n*: number of. Items’ content is abbreviated with a keyword, and the ordinate gives the item number in the questionnaire. Devices are abbreviated as in Table 1. Items and devices are ordered low to high, from the easiest to the most difficult to endorse. Extreme participants are not shown.

**Table 1 ijerph-20-01036-t001:** Number of respondents and questionnaires for each class of assistive devices.

	Patients			Caregivers			Whole Sample		
	Resp	QRE		Resp	QRE		Resp	QRE	
	*n*	*n*	%	*n*	*n*	%	*n*	*n*	%
Mobility	72	111	60.3	30	51	77.3	102	162	64.8
Seating	28	34	18.5	6	8	12.1	34	42	16.8
Furnishings	11	20	10.9	3	4	6.1	14	24	9.6
Communication	8	13	7.1	3	3	4.5	11	16	6.4
Prostheses	5	6	3.3	0	0	0.0	5	6	2.4
		184	100.0		66	100.0		250	100.0

Mobility: aids for personal mobility; seating: seating aids; furnishings: home furnishings; communication: communication aids; prostheses: lower limb prostheses. Resp: respondents. QRE: questionnaire. *n*: number of respondents or number of questionnaires. %: percentage of questionnaires referred to the total number. Counts (and percentages) are given separately for patients and caregivers. Last row: total number of questionnaires. 250 questionnaires were collected in total, 184 from 79 patients and 66 from 32 caregivers. Each participant could contribute to multiple device classes. For this reason, no respondent total is provided in the last row, and no percentage has been calculated for respondents.

**Table 2 ijerph-20-01036-t002:** Structure of the QUEST 2.0–device categories.

Categories	Count	%	Average Measures	Andrich Thresholds	
				Calibration	S.E.
1. not satisfied at all	49	3	0.13	-	-
2. not very satisfied	81	5	0.28	−0.34	0.16
3. more or less satisfied	177	10	0.70	−0.28	0.1
4. quite satisfied	401	24	1.09	0.09	0.07
5. very satisfied	982	58	1.77	0.54	0.06

Categories: categories scores and description. Count (%): number (and percentage) of observations of this category used in the analysis for parameter estimations (extreme observations excluded). Average measures: mean measures of the persons who chose the category score (extreme persons excluded; person measures are referenced to the item’s and device’s calibration). Calibration: measures of the Andrich thresholds. S.E.: standard error. The threshold between categories 1 and 2 is reported on the second row, and the remaining thresholds consequently. Average measures, calibrations, and S.E. are in logits.

**Table 3 ijerph-20-01036-t003:** Calibration and fit to the model of the QUEST 2.0–device.

		Calibration	S.E.	Infit		Outfit	
				MNSQ	ZSTD	MNSQ	ZSTD
**Items**	1. dimensions	0.06	0.08	0.94	−0.45	0.92	−0.47
	2. weight	0.36	0.08	1.04	0.42	1.17	1.17
	3. ease in adjusting	0.15	0.08	1.04	0.41	1.28	1.72
	4. safe and secure	−0.31	0.10	0.97	−0.19	0.84	−0.89
	5. durability	−0.39	0.10	0.88	−0.83	0.94	−0.26
	6. easy to use	0.09	0.08	1.08	0.74	0.98	−0.10
	7. comfortable	0.09	0.08	1.05	0.43	0.99	0.00
	8. effective	−0.06	0.09	1.10	0.80	0.91	−0.53
**Devices**	Mobility	0.37	0.04	0.96	−0.76	0.91	−1.29
	Seating	−0.28	0.10	1.19	1.36	0.99	−0.02
	Furnishings (*)	−0.17	0.12	1.43	2.33	1.59	2.45
	Communication	−1.18	0.10	1.21	1.40	1.32	1.75
	Prostheses	1.26	0.14	0.60	−2.14	0.59	−2.04

Calibration and fit to the model are reported for items and devices (upper and lower table, respectively). Calibration: items and devices measures. S.E.: standard error. MNSQ: mean square; ZSTD: z-standardised. Calibrations and S.E. are given in logit. Items are abbreviated by their number and a keyword; devices are shortened by a keyword only. The class of Furnishing devices (*) did not fit the model of Rasch. Devices are abbreviated as in Table 1.

## Data Availability

The data presented in this study are available on request from the corresponding author.

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
