# Peer review of "The Rasch Analysis Shows Poor Construct Validity and Low Reliability of the Quebec User Evaluation of Satisfaction with Assistive Technology 2.0 (QUEST 2.0) Questionnaire"

_ijerph, 2023, doi:10.3390/ijerph20021036_

Round 1

Reviewer 1 Report

The paper is interesting and very carefully written. The only thing I miss is selection of patients and incompleteness of results and loss of patients from the study (attrition bias).

Reviewer 2 Report

The manuscript studies the psychometric validity and reliability
of the eight-item Quebec User Evaluation of Satisfaction with assistive
Technology 2.0 (QUEST 2.0) questionnaire, through the analysis of a dataset collected from an administration of the questionare, analyzed by the Rasch model.In particular, 250 questionnaires were collected from 16
79 patients and 32 caregivers, where one QUEST was completed for each assistive device, among five assistive device types.

The manuscript tends to refer to quantiative methodology in more qualitative terms, while giving citations for readers who want to learn more details about these methodologies used. As such, the manuscript may be viewed as describing methodologies using only terms which may appear as jargon terms to those unfamiliar with Rasch model analysis or the field of Rasch analysis. Therefore, it would be nice if the manuscript provided a more self-contained description of these methodologies, such as the Rasch model definition, the Infit and Outfit fit statistics, and so on.

The analyses was done and reported well. One key finding was that "The device classes had different calibrations (range: -1.18 to 1.26 logits), and item 3 functioned differently in patients and caregivers". It still seems possible to argue that through the facets model, person "ability" estimates can be corrected by having the model control for differences between the device classes and the difference in the item 3 difficulty for patient versus caregivers, in analogy to regression analysis, where the estimated effect of one predictor controls for the effects of all other predictors in the model. (In fact the facets model can be viewed as an ordinal logistic regression model where the logit link is defined according to the adjacent-category function.

For these data, a possible alternative to the facets model is a "random item" Rasch model, where the item parameters are allowed to vary across the individual respondents to the Quebec User Evaluation of Satisfaction with assistive Technology 2.0 (QUEST 2.0) questionnaire (see an article published in Psychometrika, entitled "Random Item IRT").

Also, the introduction section of the article refers to the importance of the Rasch model in that it outputs Rasch person measures on an interval scale of measurement, and these measures can then be analyzed by parametric statistics. Norman Cliff some time ago argued that person scores, while they are measurable on an ordinal scale, can still be useful in that they enable the ordering of persons, and can be analyzed by rank-based hypothesis tests which only require that the scores are measured on an ordinal scale.
